# Harnessing Chlorophyll Fluorescence for Phenotyping Analysis of Wild and Cultivated Tomato for High Photochemical Efficiency under Water Deficit for Climate Change Resilience

Ilektra Sperdouli [1], Ifigeneia Mellidou [1] and Michael Moustakas [2,*]

1 Institute of Plant Breeding and Genetic Resources, Hellenic Agricultural Organization-Demeter (ELGO-Dimitra), Thermi, GR-57001 Thessaloniki, Greece; ilektras@bio.auth.gr (I.S.); imellidou@ipgrb.gr (I.M.)
2 Department of Botany, Aristotle University of Thessaloniki, GR-54124 Thessaloniki, Greece
* Correspondence: moustak@bio.auth.gr

**Abstract:** Fluctuations of the weather conditions, due to global climate change, greatly influence plant growth and development, eventually affecting crop yield and quality, but also plant survival. Since water shortage is one of the key risks for the future of agriculture, exploring the capability of crop species to grow with limited water is therefore fundamental. By using chlorophyll fluorescence analysis, we evaluated the responses of wild tomato accession *Solanum pennellii* LA0716, *Solanum lycopersicum* cv. M82, the introgression line IL12-4 (from cv. M82 X LA0716), and the Greek tomato cultivars cv. Santorini and cv. Zakinthos, to moderate drought stress (MoDS) and severe drought stress (SDS), in order to identify the minimum irrigation level for efficient photosynthetic performance. Agronomic traits (plant height, number of leaves and root/shoot biomass), relative water content (RWC), and lipid peroxidation, were also measured. Under almost 50% deficit irrigation, *S. pennellii* exhibited an enhanced photosynthetic function by displaying a hormetic response of electron transport rate (ETR), due to an increased fraction of open reaction centers, it is suggested to be activated by the low increase of reactive oxygen species (ROS). A low increase of ROS is regarded to be beneficial by stimulating defense responses and also triggering a more oxidized redox state of quinone A ($Q_A$), corresponding in *S. pennellii* under 50% deficit irrigation, to the lowest stomatal opening, resulting in reduction of water loss. *Solanum pennellii* was the most tolerant to drought, as it was expected, and could manage to have an adequate photochemical function with almost 30% water regime of well-watered plants. With 50% deficit irrigation, cv. M82 and cv. Santorini did not show any difference in photochemical efficiency to control plants and are recommended to be cultivated under deficit irrigation as an effective strategy to enhance agricultural sustainability under a global climate change. We conclude that instead of the previously used $Fv/Fm$ ratio, the redox state of $Q_A$, as it can be estimated by the chlorophyll fluorescence parameter $1 - q_L$, is a better indicator to evaluate photosynthetic efficiency and select drought tolerant cultivars under deficit irrigation.

**Keywords:** non-photochemical quenching (NPQ); moderate drought stress; severe drought stress; redox state; lipid peroxidation; singlet oxygen ($^1O_2$); hormesis; reactive oxygen species (ROS); *Solanum lycopersicum*; *Solanum pennellii*

## 1. Introduction

As a consequence of global climate change the frequency, intensity, and duration of drought is increasing and has now reached an alarming level [1–3]. Drought stress is the key factor among all environmental situations correlated with the forecast effects of climate change that will harmfully impact global crop production [4,5]. Drought stress results from below-normal precipitation, frequently combined with warm temperatures, triggering widespread damage to plants and increased risk of wildfires [2]. Although water deficit is the main cause for drought stress, increases in evapotranspiration under a

warming climate are considered as the main cause for the extensive drying under global warming [2]. Water shortage impairs osmotic adjustment by plants causing loss of turgor, impairs cell division, elongation and differentiation, and harms plants' photosynthetic rates and growth, disturbs energy distribution balance, and ultimately reduces the productivity of plants [6–8]. Drought stress accelerates leaf senescence [7,9], even a short term, resulting in crucial annual losses of crop yields that affect food security [6,10], especially when it occurs at the reproductive stage [11].

Drought-sensitive crops like the cultivated tomato (*Solanum lycopersicum* L.) are especially susceptible to impaired water availability due to climate change. *Solanum lycopersicum*, is an annual species belonging to the family Solanaceae that originated in South America and is now grown worldwide as the most popular homegrown vegetable for its edible fruits [12]. Under drought stress, *S. lycopersicum* shows decreased water and osmotic potential, resulting in low leaf relative water content [13]. As water deficit increases, stomatal openings are reduced, causing stomatal conductance and transpiration to decrease, interrupting energy dissipation and increasing leaf temperature [14].

Drought stress and high temperature significantly disturb plant productivity, which is associated mainly with the decrease of photosynthetic activity, that may be motivated both by stomatal and non-stomatal effects, which are not completely known nor understood [15–18]. Drought stress reduces photosynthesis by decreasing $CO_2$ availability through increased resistance to $CO_2$ diffusion from stomata, disturbs either biochemical, photochemical or both, activity and increases leaf membrane lipid peroxidation [19–25]. Under drought stress the absorbed light energy exceeds what can be used for photochemistry and thus excess accumulation of reactive oxygen species (ROS) occurs, that can damage the chloroplast, with photosystem II (PSII) being particularly exposed to damage [25–29]. However, overexcitation of PSII can be prevented by dissipation of excess excitation energy as heat, a procedure that is termed non-photochemical quenching (NPQ), and classically is estimated by chlorophyll a fluorescence analysis [25,30–32].

Deficit irrigation, specifically using less water than the plant requires, is proposed as an efficient approach for producing environmentally sustainable food [33]. Producing fruits and vegetables by deficit irrigation, apart from being a water-saving strategy, has become an important agronomic tradition to regulate many fruit quality variables, such as size, flavor, nutrition, and firmness [8]. Under water deficit conditions important primary and secondary metabolites that are essential for human health increase in plants [34,35]. Among these are primary metabolites, such as soluble sugars and organic acids, and secondary metabolites, e.g., anthocyanins and flavonoids, lycopene, vitamin C, and β-carotene [8,25,36]. These substances regulate nutrition and flavor of fruits and are consumers' preferences [8,37,38].

Climate change is rapidly turning into a climate crisis with unpredictable consequences for agricultural production. Since water shortage is one of the major risks for the future of agriculture and the global population, evaluating and exploring the capability of crop species to grow with limited water availability is therefore essential [36]. The development of ideas, methods, and knowhow associated with solutions to the global challenge of optimizing crop performance under water-limited conditions is a high priority research issue in order to cope with rapid climate change [39]. Plant phenotyping is an evolving area of science acquiring plant traits related to biomass development and yield, for resistance to environmental stresses associated with climate change, in a non-invasive and high throughput approach [40]. Currently, non-destructive phenotyping technologies, like chlorophyll fluorescence analysis, are of great importance as they allow predicting of plant performance under suboptimum [41] or optimum growth conditions [42], and for the assessment of photosynthetic tolerance mechanisms to biotic [43–45] and abiotic stresses [46–51], including drought stress [18,24]. Chlorophyll fluorescence analysis represents a non-destructive method that can be applied repeatedly, on the leaf- or whole-plant level, during plant growth for screening different crops for plant tolerance to several stresses and nutritional requirements [40,52–56].

The aim of this work was to evaluate responses of wild (*Solanum pennellii*) and cultivated tomato (*Solanum lycopersicum*) to deficit irrigation, and by using chlorophyll fluorescence analysis to identify the minimum irrigation level for efficient photosynthetic performance.

## 2. Materials and Methods

### 2.1. Plant Material and Growth Conditions

Seeds of the wild accession *Solanum pennellii* Correll, LA0716 (drought tolerant), *Solanum lycopersicum* L., cv. M82 (LA3475), and the introgression line IL12-4 (LA4102) from cv. M82 (*S. lycopersicum*) X LA0716 (*S. pennellii*), using M82 as the female parent, were kindly provided by the UC Davis/CM Rick Tomato Genetics Resource Center at the University of California (Davis). *S. pennellii* (LA0716) is a homozygous, self-fertile indeterminate accession from Peru with green fruits, while cv. M82 (LA3475) is a determinate, red-fruited tomato used for processing. The Greek tomato cultivars, cv. Santorini and cv. Zakinthos, that were also included in this study, were obtained from the Hellenic Agricultural Organization—Demeter (HAO-DEMETER) germplasm collection. The cv. Santorini produces small fruits, Marmande type with high vitamin C [57], while cv. Zakinthos produces large fruits with excellent taste. Although there are no reported results on the responses of these cultivars to drought stress, cv. Santorini is regarded as relatively drought-tolerant, as it grows on the volcanic island of Santorini (Greece).

All plants were grown in pots filled with peat (Terraplant, Compo) in a greenhouse at the Institute of Plant Breeding and Genetic Resources, (ELGO-Dimitra), Thermi, Thessaloniki, under 16/8 h photoperiod and temperature 22 °C $\pm$ 3 °C (day)/18 $\pm$ 3 °C (night).

### 2.2. Water-Deficit Treatments

In our experimental design we used plants from three watering regimes, determined by preliminary experiments, as follows: well-watered plants (control, 75 $\pm$ 2% of field water capacity); moderate drought stressed (MoDS), with soil water content 55 $\pm$ 5% of well-watered plants, whose watering was stopped four days before sampling; and severe drought stressed plants (SDS), with soil water content 32 $\pm$ 5% of well-watered plants, whose watering was stopped ten days before sampling. Water deficit was imposed on 5-week-old plants at the seedling stage by withholding water. These plants were transplanted to the greenhouse one week prior to this stage, i.e., at 4 weeks after germination. The control set was irrigated nonstop at regular intervals with tap water till the end of the experiment.

The irrigation regime for both well-watered and drought stressed plants was based on soil volumetric content (SWC) in $m^3\ m^{-3}$, measured by a 5TE (Decagon Devices, Pullman, WA, USA) soil moisture sensor, coupled to a ProCheck (Decagon Devices, Pullman, WA, USA) read out device [58]. Each treatment had 10 plants per genotype. All plants, i.e., control plants and plants under drought stress, were sampled on the same day for analysis.

### 2.3. Agronomic Traits Evaluation

For each treatment/genotype, plant height was recorded from the soil surface to the base of the petiole of the youngest fully expanded leaf as previously described [59,60]. Thereafter, the leaves were counted and removed for further use. In order to evaluate total biomass and relative water content (RWC), plant fresh weight (FW) and dry weight (DW) following complete drying at 70 °C were measured for belowground (roots) and aboveground tissues (stems and leaves).

### 2.4. Relative Water Content and Lipid Peroxidation Measurements

Relative water content (RWC) was assayed at the end of the experiment by cutting one leaflet from the second fully expanded leaf counting from the apex. Fresh weight of the leaflet was immediately measured and then the leaflet was immersed in a tube filled with dd-$H_2O$, following incubation under normal room temperature. After four hours, the

leaflet was wiped carefully to remove any water from the surface, and weighed to assess turgid weight (TW). In turn, the leaflet was allowed to dry in an oven for 24 h and weighed again to obtain dry weight (DW). The RWC values were calculated based on the equation: RWC (%) = (FW − DW)/(TW − DW) × 100.

Lipid peroxidation in leaf samples was assessed with 2-thiobarbituric acid (TBA) test, which determines malondialdehyde (MDA) content as the end product of lipid peroxidation [61]. Frozen leaf powder (0.20 g) was homogenized in 600 mL 0.1% (*w/v*) trichloroacetic acid (TCA) solution and proceed further as described before [60]. Absorbance of the supernatant was read at 532 nm in a Shimadzu UV-1601 spectrophotometer (Shimadzu, Kyoto, Japan). Results were expressed as μmol MDA $g^{-1}$ FW.

### 2.5. Chlorophyll Fluorescence Measurements

Chlorophyll fluorescence was measured in dark-adapted (20 min) tomato leaves, using an imaging-PAM fluorometer (Walz, Effeltrich, Germany) as described previously [62]. The minimum chlorophyll *a* fluorescence in the dark (*Fo*), the maximum chlorophyll *a* fluorescence in the dark (*Fm*), the maximum chlorophyll *a* fluorescence in the light (*Fm'*), and the steady-state photosynthesis in the light (*Fs*) were the basic chlorophyll fluorescence parameters that were measured [63]. The actinic light (AL) used for chlorophyll fluorescence measurements was 636 μmol photons $m^{-2} s^{-1}$. The minimum chlorophyll *a* fluorescence in the light (*Fo'*) was calculated using Imaging Win V2.41a software (Heinz Walz GmbH, Effeltrich, Germany) as $Fo' = Fo/(Fv/Fm + Fo/Fm')$. By using these basic chlorophyll fluorescence parameters, Win software calculated the maximum efficiency of PSII photochemistry ($Fv/Fm = (Fm − Fo)/Fm$), the actual quantum yield of PSII photochemistry ($\Phi_{PSII} = (Fm' − Fs)/Fm'$), the quantum yield of regulated non-photochemical energy loss in PSII ($\Phi_{NPQ} = Fs/Fm' − Fs/Fm$), and the quantum yield of non-regulated energy dissipated in PSII ($\Phi_{NO} = Fs/Fm$). The relative PSII electron transport rate (ETR = $\Phi_{PSII}$ × PAR × c × abs, where c is 0.5, abs is the total light absorption of the leaf taken as 0.84, and PAR is the photosynthetically active radiation, e.g., 636 μmol photons $m^{-2} s^{-1}$), the redox state of quinone A ($Q_A$) ($q_p = (Fm' − Fs)/(Fm' − Fo')$), an estimate of the fraction of open PSII reaction centers based on the "puddle" model for the photosynthetic unit, the relative excess energy at PSII (EXC = $(Fv/Fm − \Phi_{PSII})/(Fv/Fm)$) according to Bilger et al. [64], the non-photochemical quenching that reflects heat dissipation of excitation energy (NPQ = $(Fm − Fm')/Fm'$), and the redox state $Q_A$ or the fraction of open PSII reaction centers that are connected by shared antenna, that is, the so-called "lake" model ($q_L = q_p × Fo'/F_S$) [65], were also calculated.

### 2.6. Statistical Analysis

Data were analyzed following one-way analysis of variance (ANOVA). To detect statistical significance of differences between means of the treatments, the Duncan test was performed at 5% level of significance using the statistical package SPSS (version 24). Data are presented as mean values ± standard error. A linear regression analysis was also performed [34]. Experiments were repeated three times with 3–5 plants measured in each experiment under each treatment.

## 3. Results

### 3.1. Changes in Agronomic Traits and Relative Water Content under Drought Stress

The agronomic traits that were measured were plant height, the number of leaves, and the ratio root/shoot biomass. Under MoDS, plant height decreased in M82 and Zakinthos, while it remained unaffected in *S. pennellii*, Santorini and IL12-4, compared to well-watered plants (control) (Figure 1a). Under SDS, plant height decreased in all cultivars except the introgression line IL12-4, compared to control (Figure 1a). Of all plants, *S. pennellii* retained the greatest height under SDS (Figure 1a).

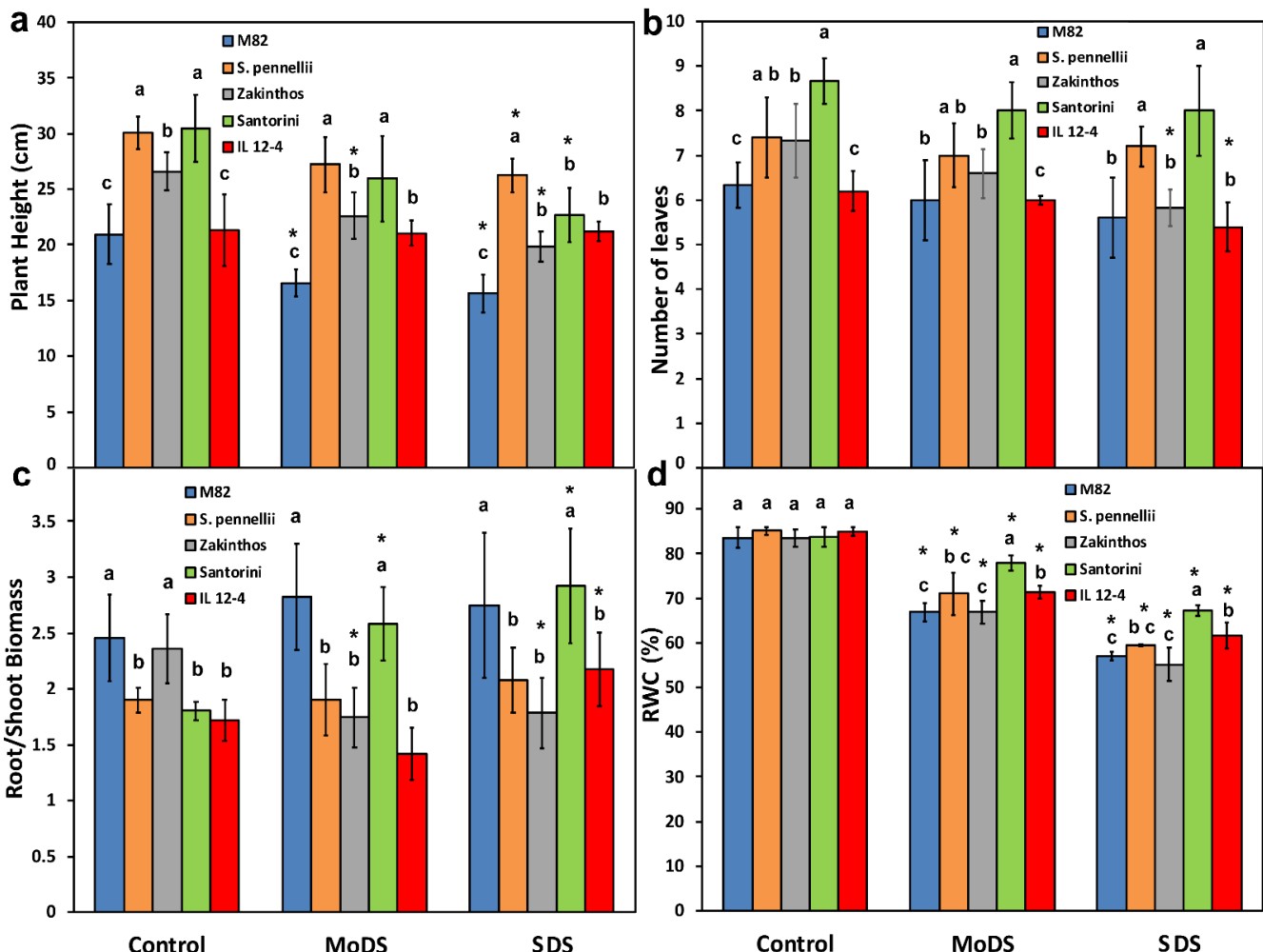

**Figure 1.** Changes in plant height (**a**), in the number of leaves (**b**), in the ratio of root to shoot biomass (**c**), and in the relative water content (RWC) (**d**), of *S. pennellii*, the introgression line IL12-4, and *S. lycopersicum* cv. M82, cv. Zakinthos, and cv. Santorini, under well-watered (control), MoDS and SDS. Error bars on columns are standard errors. Columns with different lowercase letters are statistically different (*p* < 0.05). An asterisk (*) represents a significantly (*p* < 0.05) different mean between the treatments.

The number of leaves remained unaffected under MoDS, while under SDS it was reduced only in Zakinthos and IL12-4 (Figure 1b). Of all plants under SDS, *S. pennellii* and Santorini retained the highest number of leaves, which was the same as control plants (Figure 1b).

The ratio of root/shoot biomass under MoDS increased in Santorini, while it decreased in Zakinthos (Figure 1c). There was no change in root/shoot biomass under MoDS in M82, *S. pennellii*, and IL12-4 (Figure 1c). Under SDS, root/shoot biomass decreased in Zakinthos, but increased in IL12-4 (due to a lower decrease in root biomass, data not shown) and Santorini (due to a decrease in shoot biomass without any significant change in root biomass, data not shown) (Figure 1c). Finally, there was no change in the ratio of root/shoot biomass under SDS in M82 and *S. pennellii* compared to controls (Figure 1c).

Under well-watered conditions there was no difference among all plants in the relative water content (RWC). Under both MoDS and SDS, the relative water content (RWC) decreased significantly in all plants compared to well-watered (control) plants (Figure 1d). Among all plants, Santorini retained the higher RWC under both MoDS and SDS (Figure 1d).

### 3.2. The Level of Lipid Peroxidation under Drought Stress

The level of lipid peroxidation, measured as μmol MDA g$^{-1}$ fresh weight under both MoDS and SDS increased significantly in all plants compared to their controls (Figure 2a). The lowest increase of lipid peroxidation compared to control plants, under both MoDS and SDS, was noticed in *S. pennellii* (Figure 2a).

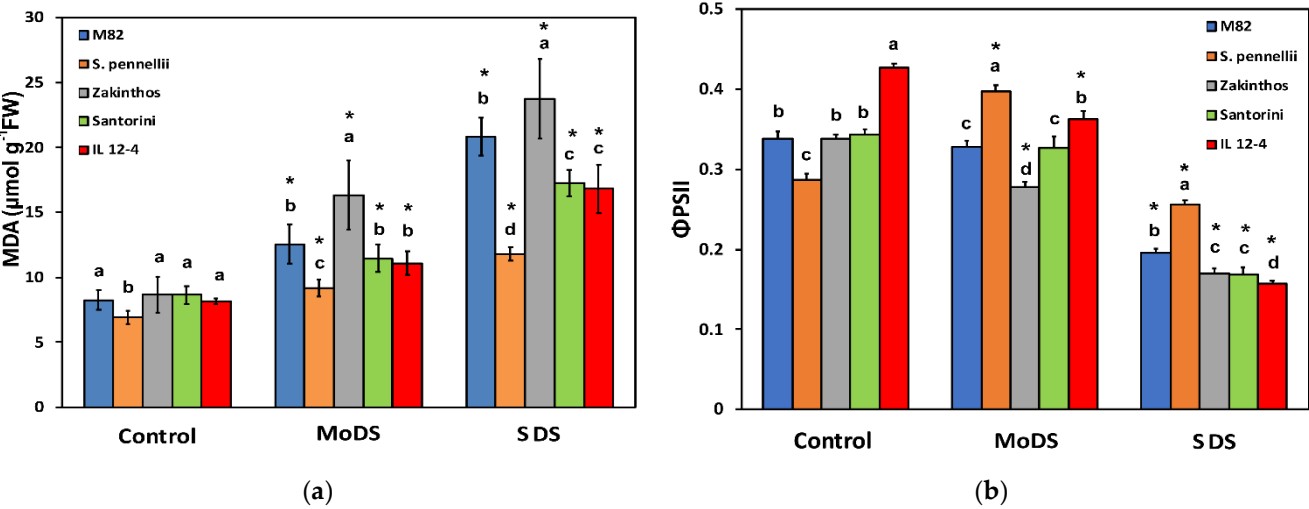

**Figure 2.** Changes in the level of lipid peroxidation, measured as μmol MDA g$^{-1}$ fresh weight (**a**), and in the effective quantum yield of photochemistry ($\Phi_{PSII}$) (**b**), of *S. pennellii*, the introgression line IL12-4, and *S. lycopersicum* cv. M82, cv. Zakinthos, and cv. Santorini, under well-watered conditions(control), MoDS, and SDS. Error bars on columns are standard errors. Columns with different lowercase letters are statistically different ($p < 0.05$). An asterisk (*) represents a significantly ($p < 0.05$) different mean between the treatments.

### 3.3. Light Energy Utilization in Photosystem II under Drought Stress

The changes in light energy utilization in PSII of the tomato cultivars and the introgression line under MoDS and SDS were estimated by measuring the chlorophyll fluorescence parameters $\Phi_{PSII}$, $\Phi_{NPQ}$, and $\Phi_{NO}$, the sum of all them equal to 1 [65].

The effective quantum yield of photochemistry ($\Phi_{PSII}$) under MoDS decreased in Zakinthos and IL12-4, but increased in *S. pennellii*, while it remained unaffected in M82 and Santorini (Figure 2b). Under SDS, $\Phi_{PSII}$ decreased in all cultivars (Figure 2b). Under both MoDS and SDS, *S. pennellii* possessed the higher $\Phi_{PSII}$ (Figure 2b).

The quantum yield of regulated non-photochemical energy loss in PSII ($\Phi_{NPQ}$) under MoDS increased in M82, Zakinthos, and IL12-4, but decreased in *S. pennellii* while it remained unaffected in Santorini compared to control plants (Figure 3a). Under SDS, $\Phi_{NPQ}$ increased in all cultivars (Figure 3a).

The quantum yield of non-regulated energy dissipated in PSII ($\Phi_{NO}$), under MoDS, it did not change in Zakinthos and Santorini while it decreased in M82, *S. pennellii*, and IL12-4, compared to their controls (Figure 3b). Under SDS, $\Phi_{NO}$ increased in Santorini and IL12-4, but decreased in M82 and *S. pennellii*, while it remained unaffected in Zakinthos, compared to well-watered plants (Figure 3b).

### 3.4. Maximum Efficiency of Photosystem II and the Fraction of Open PSII Centers

The maximum efficiency of PSII photochemistry (*Fv/Fm*), under MoDS, decreased in all plants except the introgression line IL12-4 in which it decreased under SDS (Figure 4a).

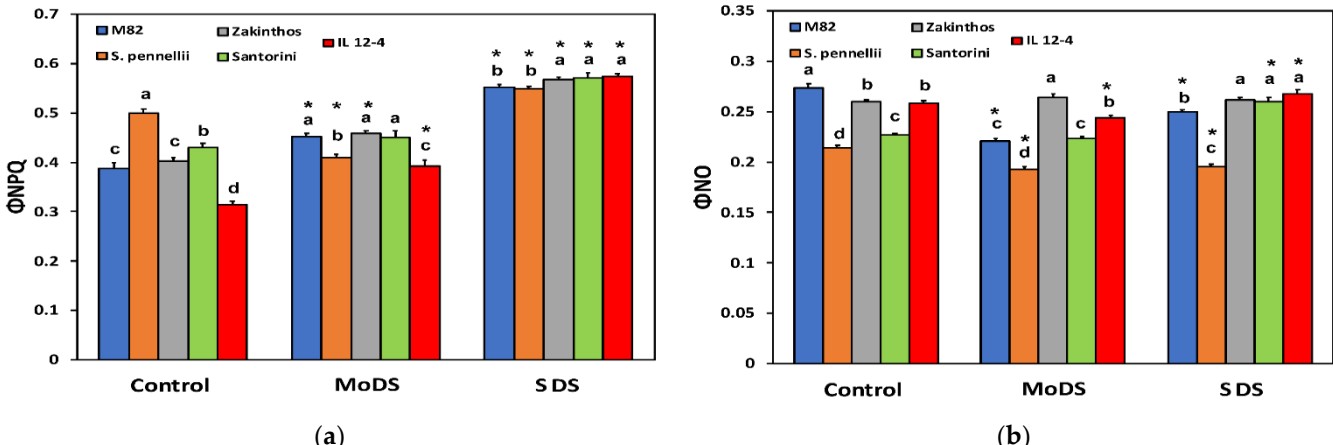

**Figure 3.** Changes in the quantum yield of regulated non-photochemical energy loss in PSII ($\Phi_{NPQ}$) (**a**), and in the quantum yield of non-regulated energy dissipated in PSII ($\Phi_{NO}$) (**b**), of *S. pennellii*, the introgression line IL12-4, and *S. lycopersicum* cv. M82, cv. Zakinthos, and cv. Santorini, under well-watered conditions (control), MoDS, and SDS. Error bars on columns are standard errors. Columns with different lowercase letters are statistically different ($p < 0.05$). An asterisk (*) represents a significantly ($p < 0.05$) different mean between the treatments.

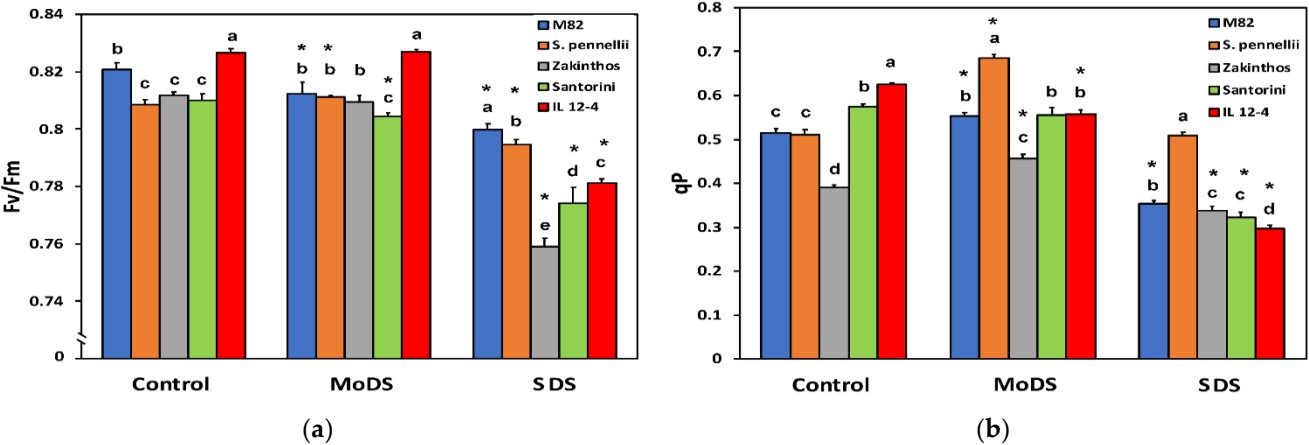

**Figure 4.** Changes in the maximum efficiency of PSII photochemistry (*Fv/Fm*) (**a**), and in the redox state of the plastoquinone pool ($q_p$), an estimate of the fraction of open PSII reaction centers (**b**), of *S. pennellii*, the introgression line IL12-4, and *S. lycopersicum* cv. M82, cv. Zakinthos, and cv. Santorini, under well-watered conditions (control), MoDS, and SDS. Error bars on columns are standard errors. Columns with different lowercase letters are statistically different ($p < 0.05$). An asterisk (*) represents a significantly ($p < 0.05$) different mean between the treatments.

The redox state of the plastoquinone pool ($q_p$), an estimate of the fraction of open PSII reaction centers, under MoDS decreased in IL12-4, but increased in M82, *S. pennellii*, and Zakinthos, while remaining unchanged in Santorini, compared to their controls (Figure 4b). Under SDS, $q_p$ decreased in all cultivars, except in *S. pennellii* in which it remained at the level of control plants (Figure 4b).

### 3.5. Heat Dissipation in Photosystem II and Electron Transport Rate

Non-photochemical quenching (NPQ), under MoDS, increased in M82, Zakinthos, and IL12-4, but decreased in *S. pennellii*, while it remained unaffected in Santorini (Figure 5a). Under SDS, NPQ increased in all cultivars with *S. pennellii* having the highest values (Figure 5a).

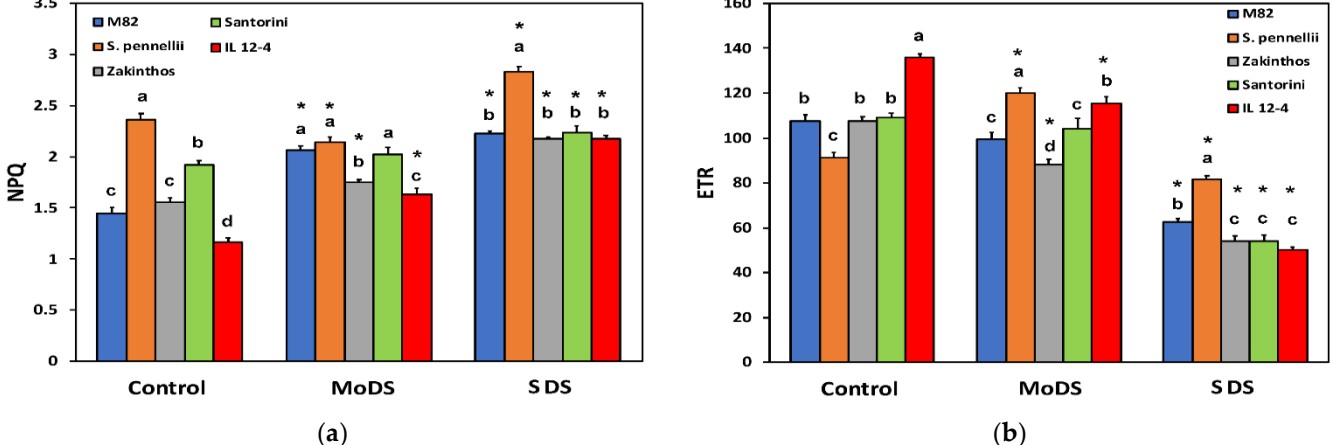

(**a**)  (**b**)

**Figure 5.** Changes in non-photochemical quenching that reflects heat dissipation of excitation energy (NPQ) (**a**), and in the relative PSII electron transport rate (ETR) (**b**), of *S. pennellii*, the introgression line IL12-4, and *S. lycopersicum* cv. M82, cv. Zakinthos, and cv. Santorini, under well-watered conditions (control), MoDS, and SDS. Error bars on columns are standard errors. Columns with different lowercase letters are statistically different ($p < 0.05$). An asterisk (*) represents a significantly ($p < 0.05$) different mean between the treatments.

The relative PSII electron transport rate (ETR), under MoDS, decreased in Zakinthos and IL12-4, but increased in *S. pennellii*, while it remained unaffected in M82 and Santorini (Figure 5b). Under SDS, ETR decreased in all cultivars, having the highest values in *S. pennellii* (Figure 5b).

### 3.6. The Redox State of Plastoquinone Pool Based on the Lake Model and the Excess Excitation Energy in Photosystem II

The redox state of $Q_A$ based on the lake model ($1 - q_L$) under MoDS, became more oxidized in M82 and *S. pennellii*, but in Zakinthos and IL12-4 it became more reduced, while it did not change in Santorini, compared to control plants (Figure 6a). Under SDS, it became more reduced in all plants except for *S. pennellii* in which it remained more oxidized compared to controls (Figure 6a).

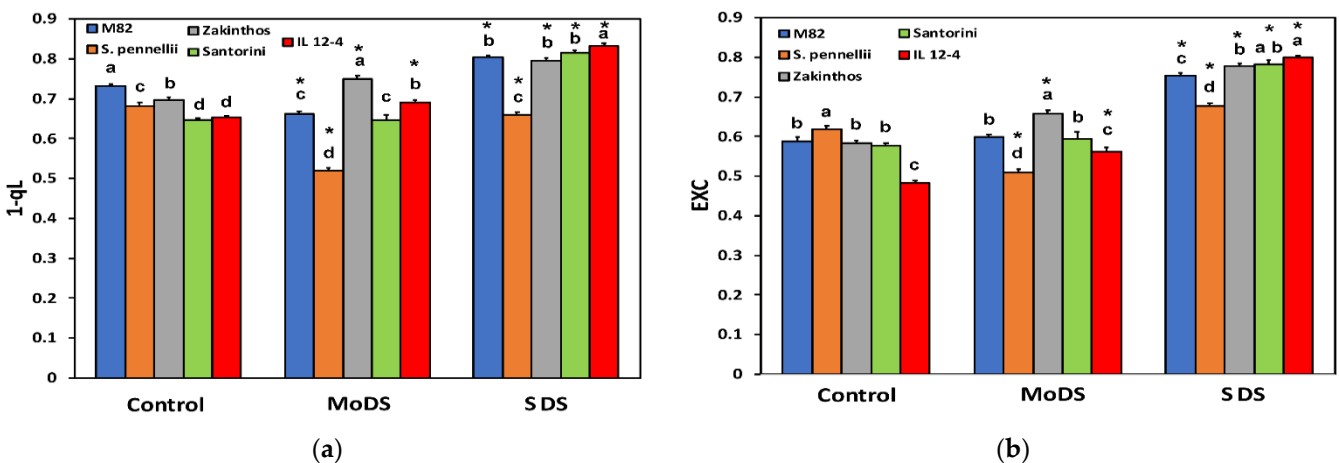

(**a**)  (**b**)

**Figure 6.** Changes in the parameter $1 - q_L$, that is the fraction of closed PSII centers based on a lake model for the photosynthetic unit (**a**), and in the relative excess energy at PSII (EXC) (**b**), of *S. pennellii*, the introgression line IL12-4, and *S. lycopersicum* cv. M82, cv. Zakinthos, and cv. Santorini, under well-watered conditions (control), MoDS, and SDS. Error bars on columns are standard errors. Columns with different lowercase letters are statistically different ($p < 0.05$). An asterisk (*) represents a significantly ($p < 0.05$) different mean between the treatments.

The relative excess energy at PSII (EXC), under MoDS, increased in Zakinthos and IL12-4, remained unaltered in Santorini and M82, but decreased in *S. pennellii* compared to control plants (Figure 6b). Under SDS, EXC increased in all plants compared to controls. However, *S. pennellii* presented the lowest excess excitation energy (Figure 6b).

*3.7. Regression Analysis between the Relative Excess Energy in Photosystem II and the Level of Lipid Peroxidation*

The level of lipid peroxidation, measured as $\mu$mol MDA g$^{-1}$ fresh weight, at the three watering regimes, well-watered (control), moderate drought stressed (MoDS), and severe drought stressed (SDS), was strongly correlated ($R^2 = 0.8869$, $p < 0.001$) to the level of excess excitation energy at PSII (EXC), at the light intensity of 636 $\mu$mol photons m$^{-2}$ s$^{-1}$, in *S. pennellii*, *S. lycopersicum* cv. M82, cv. Zakinthos, cv. Santorini, and the introgression line IL12-4 (Figure S1a).

*3.8. Regression Analysis between the Relative Excess Energy in Photosystem II and the Redox State of Photosystem II (1 - $q_L$)*

The redox state of photosystem II, based on the lake model, evaluated as 1 - $q_L$ at the light intensity of 636 $\mu$mol photons m$^{-2}$ s$^{-1}$, of control, moderate drought stressed (MoDS), and severe drought stressed (SDS), in *S. pennellii*, *S. lycopersicum* cv. M82, cv. Zakinthos, cv. Santorini, and the introgression line IL12-4, was strongly correlated ($R^2 = 0.9231$, $p < 0.001$) to the level of excess excitation energy at PSII (EXC), also measured at the light intensity of 636 $\mu$mol photons m$^{-2}$ s$^{-1}$ (Figure S1b).

## 4. Discussion

The efficient use of available water is a critical research issue to plant production in water-limited environments for climate change resilience [39]. Plant phenotyping for breeding and for precision agriculture is an urgency requiring action to mitigate rapid climate change and the demand for sustainable agriculture and increased biomass production [8,66,67]. Recent advances in sensor technology and assessment of agronomic traits, such as biomass and plant height, have been obtained that should help in understanding complex physiological processes that determine yield, at a scale that cannot be achieved by manual methods [66,68,69]. Deficit irrigation must be applied in combination with other cultural practices such as pruning, grafting, de-leafing, and fertilization, which also have a substantial impact on tomato fruit quality [8,70–72]. Although chlorophyll fluorescence is a dominant non-destructive technique for probing photosynthetic tolerance to drought stress [25], and has frequently been used as a method for drought tolerance screening [50,51,73], it has not yet been totally implemented in physiological breeding [73] and for evaluation of the minimum irrigation levels for efficient photosynthetic performance.

The maximum efficiency of PSII photochemistry (*Fv/Fm*) is among the chlorophyll fluorescence parameters mostly used to evaluate drought stress impact on plants and for selection of drought-tolerant cultivars [25,74]. In our experiment, the *Fv/Fm* ratio that was measured in dark-adapted (20 min) leaves was found to decrease under MoDS in all cultivars, but to remain unaffected in the introgression line IL12-4 which decreased only under SDS (Figure 4a). However, the use of *Fv/Fm* as an efficient indicator has been frequently questioned [25,74–76], and recently it was recommended that the *Fv/Fm* parameter must not to be related to the efficiency of PSII photochemistry [77,78]. In contrast, the pattern of the redox state of quinone A ($Q_A$) has been shown to be a good indicator to probe photosynthetic efficiency and to determine the impact of abiotic or biotic stress on photosynthesis [25,43,79]. In accordance with this, in our experiment, the pattern of the redox state of $Q_A$ (Figure 4b) was found to be related to the pattern of MDA (Figure 2a), while the pattern of *Fv/Fm* (Figure 4a) under MoDS or SDS, did not match that of lipid membrane peroxidation (Figure 2a), or the patterns of agronomic traits (Figure 1a–c). MDA, the final product of cell membrane lipid peroxidation is considered a reliable indicator of membrane system injury [22,80].

MDA contents revealed a significant increase of lipid peroxidation under both MoDS and SDS compared to well-watered (control) plants (Figure 2a). However, *S. pennellii* showed the lowest increase in lipid peroxidation under drought stress, and could be suggested as being the most tolerant to water deficit. In agreement with this pattern, under both MoDS and SDS, *S. pennellii* showed the highest fraction of open reaction centers compared to all cultivars, and under MoDS this fraction was even higher than well-watered (control) plants, while under SDS it was at the same level as control plants (Figure 4b). In other words, *S. pennellii* plants under MoDS showed a higher oxidized redox state of $Q_A$ compared to their control plants, while under SDS, the same redox state of $Q_A$ to control plants (Figure 4b). We may suggest that the low increase of lipid peroxidation in *S. pennellii* under MoDS is possible due to an increased production of ROS, occurring under drought stress [25–29], which can result to an acclimatory response [81–86] and sometimes in a hormetic response [44,87,88]. Hormesis is described as the stimulation effect of low doses or short time exposures and a high dose or longer duration inhibition, to a variety of stressors; being a widespread phenomenon in nature [44,87,89–91]. Hormesis represents an "over-compensation" response to a disruption in homeostasis and is considered a fundamental evolutionary adaptive strategy [92,93].

The increased $\Phi_{PSII}$ in *S. pennellii* plants under MoDS (Figure 2b) was due to the decreased NPQ (Figure 5a) that resulted in increased electron transport rate (ETR) (Figure 5b). This hormetic response of ETR in *S. pennellii* plants under MoDS, was due to an increased fraction of open reaction centers (Figure 4b). Under SDS, a decreased ETR (Figure 5b) followed the decreased fraction of open reaction centers (Figure 4b). A low increase in ROS level is regarded to be beneficial by activating defense responses [84,85,87], also triggering an increase in the fraction of open PSII reaction centers [85], as observed in *S. pennellii* under MoDS, compared to controls (Figure 4b), indicating an enhanced PSII functionality [85].

Chloroplasts, through the process of photosynthesis, play a central role as redox sensors of environmental situations and elicit acclimatory or stress defense responses [94–96]. The chloroplast redox state has an important impact on plant growth, development, and defense, that goes beyond its role in primary metabolism [97]. The redox state of $Q_A$ is also regarded as a sensor of the energy imbalance under environmental stress conditions [98,99]. Accordingly, *S. pennellii* plants under both MoDS and SDS showed lower excess energy at PSII (EXC) compared to all cultivars (Figure 6b). Over-reduction of the electron transport can severely damage the chloroplast and the cell [100]. The decreased capacity, of Zakinthos and IL12-4 to keep quinone ($Q_A$) oxidized under MoDS and SDS (Figure 4b or Figure 6a), was accompanied by an excess excitation energy at PSII (Figure 6b). High excess excitation energy and therefore an imbalance between energy supply and demand results in increased ROS production [43,87,101]. In accordance, the relative excess energy in PSII was strongly correlated to both the level of lipid peroxidation (regression coefficient $R^2 = 0.8869$) and the redox state of PSII ($1 - q_L$) (regression coefficient $R^2 = 0.9231$).

An increased ROS production, such as, hydrogen peroxide ($H_2O_2$), singlet oxygen ($^1O_2$), superoxide ($O_2^-$), and hydroxyl radical ($HO\cdot$) under diverse environmental stressors [82,102–107] and in response to drought stress [25] can cause cellular damage by oxidation of DNA, proteins, and lipids and can result in oxidative stress [25,30,82,101]. Oxidative stress that is commonly assessed by malondialdehyde (MDA) content, a marker of lipid peroxidation [80,108], was found among all tomato cultivars, under both MoDS and SDS, increasing more in Zakinthos, in which a high increase of non-regulated energy dissipated in PSII ($\Phi_{NO}$) was observed. $\Phi_{NO}$ is comprises of chlorophyll fluorescence interior conversions and intersystem crossing, which leads to the generation of $^1O_2$ via the triplet state of chlorophyll ($^3chl^*$) [30,63,82,104,109–114]. Therefore, the increased $\Phi_{NO}$ values in Zakinthos (Figure 3b) suggest a higher level of $^1O_2$ formation under drought stress, and the lower $\Phi_{NO}$ values in *S. pennellii* plants under both MoDS and SDS, suggest a lower level of $^1O_2$ formation (Figure 3b).

Photosystem II of higher plants, under environmental stress conditions, is protected against excess energy supply that leads to ROS production by thermal dissipation of

the excess excitation energy, a process that can be perceived through non-photochemical quenching (NPQ) of chlorophyll fluorescence [115,116]. NPQ is considered as the main photoprotective process that dissipates excess light energy as heat and protects photosynthesis under drought stress conditions, preventing the formation of ROS [25,30,32,117–120]. Under SDS, the induction of the NPQ mechanism in *S. pennellii* plants to dissipate excessive excited energy as heat (Figure 5a) downregulated the light energy utilized in photochemistry (Figure 2b), decreasing ETR (Figure 5b), but resulting in avoidance of the harmful generation of $^1O_2$ (Figure 3b), which can damage the photosynthetic apparatus [30,58,121]. The photoprotective mechanism of NPQ can be regarded as efficient, under abiotic or biotic stress conditions, if it can retain the same redox state of $Q_A$ as in control conditions [41,115]. Thus, the induction of the NPQ mechanism in *S. pennellii* plants under SDS was efficient enough to maintain the same redox state of $Q_A$ as in control plants (Figure 4b), and to prevent generation of the harmful $^1O_2$ (Figure 3b). NPQ is involved in the mechanism of plant acclimation to biotic or abiotic stress and has also been suggested to be a major component of the systemic acquired resistance [43,44,85,122–124].

Changes in the redox state of $Q_A$, as estimated by the chlorophyll fluorescence parameter $1 - q_L$ [65] act as a signal to the stomatal guard cells [99]. Accordingly, the higher oxidized $Q_A$ pool under MoDS in *S. pennellii* plants of all cultivars (Figure 6a) corresponds to the lowest stomatal opening among all cultivars. Coherent with this hypothesis, $1 - q_L$ was strongly and linearly correlated to stomatal conductance in tobacco with modified levels of the photosystem II subunit S (PsbS) [125]. The protein PsbS plays an essential role in triggering NPQ responses, involved in the photoprotective mechanism to dissipate over-excitation harmlessly [45,126]. Increased PsbS expression is associated with increased levels of NPQ [45,126]. A greater increase in NPQ, which is a characteristic of drought-tolerant cultivars [127], was found in *S. pennellii*, indicating that the drought-tolerant cultivar managed over-excitation of PSII by harmless heat dissipation via the photo-protective mechanism of NPQ [127].

Plants with increased PsbS expression, that is, with increased levels of NPQ, show less stomatal opening in response to light, resulting in a 25% reduction in water loss per $CO_2$ assimilated under field conditions [128]. It appears that the increased levels of NPQ under SDS in *S. pennellii* plants among all tomato cultivars examined, caused less stomatal opening. It has been suggested that plants can counteract the effects of a decrease in stomatal conductance by increasing photosynthetic function, thereby limiting the negative feedback on biomass productivity [125]. A confirmation of this is the increased ETR in *S. pennellii* plants under SDS, compared to all tomato cultivars (Figure 5b), which might have contributed to the increased plant height observed (Figure 1a). It seems, according to resent evidence, that stomatal movement is controlled by the redox state of the plastoquinone (PQ) pool instead of the Calvin–Benson cycle or the rate of $CO_2$ assimilation [129].

Maintaining appropriate ROS scavenging capacity is critical for sustaining plant growth and development in response to drought stress [3,25,130]. Nevertheless, ROS are also considered to be important signaling molecules that regulate plant development and various abiotic and biotic stress responses [81,84,85,102]. An appropriate response to an environmental stressor depends mainly on how plants recognize the stress signal, and reacts to initiate a series of signalling cascades for induction of acclimation mechanisms [131].

Under optimal water regime the introgression line IL12-4 had the highest effective quantum yield of photochemistry ($\Phi_{PSII}$) of all plants, whereas the wild tomato, *S. pennellii*, had the lowest. However, under both MoDS and SDS, *S. pennellii* possessed the highest $\Phi_{PSII}$, while the lowest $\Phi_{PSII}$ under MoDS was revealed in Zakinthos and under SDS in the introgression line IL12-4 (Figure 2b).

The introgression line IL12-4 (LA4102), under optimal water regime, showed a better photochemical functioning from both its parents (*S. lycopersicum* cv. M82 and *S. pennellii* LA0716) but under MoDS the worst from *S. pennellii*, while under SDS the worst from both parents. Under both MoDS and SDS, *S. pennellii* and cv. Santorini retained the same number of leaves as their controls (Figure 1b), while under MoDS the same plant height to

controls that decreased under SDS (compared to controls), though remaining the highest of all (Figure 1a).

## 5. Conclusions

Under deficit irrigation, specifically using almost 50% less water than the control tomato plants, *S. pennellii* displayed an enhanced photosynthetic function. This was achieved despite a decreased relative leaf water content that did not influence plant height, number of leaves, or the root/shoot biomass ratio.

Under 50% less water, the tomato cultivars, M82 and Santorini, were found to have no difference in photochemical efficiency compared to control plants and thus can be regarded as tolerant to water deficit. Photosystem II efficiency of the wild tomato *S. pennellii* was the most tolerant to drought, as was expected, and could manage adequate photochemical function with almost 30% water regime of well-watered plants.

From the evaluation of our results, we can conclude that chlorophyll *a* fluorescence analysis is suitable for photosynthetic efficiency estimation, permitting probing and elucidating of tomato cultivar responses to drought stress. Among the chlorophyll fluorescence parameters examined, the redox state of quinone A ($Q_A$) was found to be a good indicator to reveal short- or long-term stress impacts on the mechanisms of PSII functionality. Chlorophyll fluorescence is a promising phenotyping technique that allows early and quick detection of drought stress effects and thus it can be used for drought tolerance screening in physiological breeding and for the evaluation of the minimum irrigation levels for efficient photosynthetic performance.

Based on our results, we conclude that the frequently used *Fv/Fm* ratio is not a proper indicator for selection of drought-tolerant cultivars, since it has also recently been shown not to be related to the efficiency of PSII photochemistry [77,78], but instead, the redox state of $Q_A$, as it can be estimated by the chlorophyll fluorescence parameter 1 - $q_L$, is proposed as a good indicator to evaluate photosynthetic efficiency and to select drought tolerant cultivars under deficit irrigation.

Future extended experiments with more cultivars and growth stages based on the evaluation of the redox state of $Q_A$ can be applied to agricultural systems to reduce irrigation and increase productivity at far lower costs, helping to enhance agricultural sustainability under global climate change.

**Supplementary Materials:** The following is available online at https://www.mdpi.com/article/10.3390/cli9110154/s1, Figure S1: The relationship between the excess excitation energy at PSII with the level of lipid peroxidation, and the parameter 1 - $q_L$.

**Author Contributions:** Conceptualization, I.S., I.M. and M.M.; methodology, I.S. and I.M.; software, I.M.; validation, I.S., I.M. and M.M.; formal analysis, I.S. and I.M.; investigation, I.S. and I.M.; resources, I.M. and M.M.; data curation, I.S. and I.M.; writing—original draft preparation, I.S., I.M. and M.M.; writing—review and editing, I.S., I.M. and M.M.; supervision, M.M.; project administration, I.S., I.M. and M.M. All authors have read and agreed to the published version of the manuscript.

**Funding:** This research received no external funding.

**Institutional Review Board Statement:** Not applicable.

**Informed Consent Statement:** Not applicable.

**Data Availability Statement:** The data presented in this study are available in this article.

**Conflicts of Interest:** The authors declare no conflict of interest.

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
