# Peer review of "Harnessing Chlorophyll Fluorescence for Phenotyping Analysis of Wild and Cultivated Tomato for High Photochemical Efficiency under Water Deficit for Climate Change Resilience"

_climate, doi:10.3390/cli9110154_

Round 1

Reviewer 1 Report

Harnessing Chlorophyll Fluorescence for Phenotyping Analysis of Wild and Cultivated Tomato for High Photochemical Efficiency under Water Deficit for Climate Change Resilience

This is a well-structured, easily comprehended ms., describing the findings of what appears to be a competent piece of research. The methods and equipment used seem to be appropriate, given the experimentation requirements and questions at hand. I do, however, have a number of key questions and reservations, which I feel need to be carefully addressed. In particular, some of my comments draw question as to the uniqueness and strength of the reported outcomes and implications given in the paper; these require particular attention. Some of the comments that need to be addressed are based around climate understanding and are thus germane to the periodical of choice, title of the paper, and ms. narrative.

Specific comments, by line(s)

Line 11

“Global climate change greatly influences plant growth and development, eventually affecting crop yield and quality”

  1. This needs rephrasing. Global climate change is not the ultimate driver but rather climate variables, via weather. The outcomes of climate change not only influences crop yield and quality (morbidity), but also survival (mortality).

Lines 26-28

“With almost 50% less water of well-watered plants, cv. Μ82 and cv. Santorini can be regarded, from all tomato cultivars examined, as water deficit-tolerant, with no difference in photochemical efficiency to control plants.”

  1. This doesn’t make any sense. Only two tomato cultivars were examined; and two were water deficit-tolerant. Does this study include enough cv. replicates to make the overarching claims made in the ms.
  2. What about 30% less water scenario. This is not reported in the abstract and doesn’t get much coverage in the ms.
  3. 50% may be the optimum water content, and thus not a deficit. See comment directly below.

Lines 28-29

“Thus, we can propose deficit irrigation as an effective strategy to enhance agricultural sustainability under a global climate change.”

  1. I don’t think this is actually deficit irrigation. Most irrigation schemes do not aim for field capacity, but rather what the plant requires for healthy growth and productivity.

Line 34

“As a consequence of global climate change the frequency, intensity, and duration of drought is increasing and has now reach[ed] to an alarming level [1-3]”

  1. Generally the English usage in the paper is very good, but please check ms. carefully and update accordingly.

Lines 37-39

“Drought stress results from below-normal precipitation, frequently combined with warm temperatures, triggering widespread damages to plants and increased risk of wildfires [2].”

  1. There are also many other factors, as you have alluded to in lines 66-68. These should be brought together under the same section, and other factors itemized and briefly discussed.

Lines 46-47

“Drought-sensitive crops like the cultivated tomato (Solanum lycopersicum L.) are especially susceptible to impaired water availability due to climate change.”

  1. Why? Aren’t most crops susceptible?
  2. The emphasis is on water availability, but photosynthetic efficiency/performance is also strongly influence by temperatures.

Lines 47-50

“Solanum lycopersicum, is an annual species belonging to the family Solanaceae that originated in South America and now grown worldwide, as the most popular homegrown vegetable, for its edible fruits.”

  1. Please add reference.

Lines 66-68

“Although water deficit is the main cause for drought stress, increases in evapotranspiration under a warming climate are considered as the main cause for the extensive 67 drying under global warming [2]”

Please see comment above for lines 37-39.

Lines 68-70, and lines 70-73

“On the other hand, deficit irrigation, specifically using less water than the plant requirements, is proposed as an effective strategy for producing environmentally sustainable food [33].”

“Producing fruits and vegetables by deficit irrigation will contribute optimistically to the future of the planet and apart from being a water-saving strategy, deficit irrigation has become an important agronomic tradition to regulate many fruit quality variables [8].

  1. Most of the world’s food in grown under rain-fed conditions. Surely non-irrigation is the key, particularly given the carbon requirement needed to undertake irrigation, and the increasing demand for drinking water.
  2. “…will contribute optimistically to the future of the planet…”

I don’t think so. Reducing irrigation will help, but overall the impact would be negative’ unless the water was reclaimed using carbon-free technology.

  1. I think it’s also important to remember that much of the world’s commercial tomato production uses hydroponics, with already carefully monitored water usage.

Lines 115-117.

“All plants were grown in pots filled with peat (Terraplant, Compo) in a greenhouse at the Institute of Plant Breeding and Genetic Resources, (ELGO-Dimitra), Thermi, Thessaloniki, under 16h/8h photoperiod and temperature 22°C ± 3°C (day)/18 ± 3°C (night).”

  1. What were the growth stage used?
  2. Over what time period?
  3. Were all life stages included, and most importantly the fruiting stage?

Lines 119-124

“In our experimental design we used plants from three watering regimes, determined by preliminary experiments, as follows: well-watered plants (control, 75 ± 2 % of field water capacity), moderate drought stressed (MoDS), with soil water status 55 ± 5% 121 of well-watered plants, whose watering stopped four days before sampling, and severe 122 drought stressed plants (SDS), with soil water status 32 ± 5% of well-watered plants,  whose watering stopped ten days before sampling.

  1. In addition to % of field capacity, also provide the actual readings from the probe/sensor in m3 m−3.
  2. The amount of water available to the plant also depends on the soil. Ideally, soil water potential should have been measured with a suitable Decagon probe.
  3. Ideally a further water deficit trial should have been included, as 30% deficit still provides the plant with sufficient water. This is borne out by the results.

Lines 331-333

“The maximum efficiency of PSII photochemistry (Fv/Fm) is among the chlorophyll fluorescence parameters mostly used to evaluate drought stress impact on plants and for selection of drought-tolerant varieties [25,74].”

  1. Cultivars, not varieties.

Lines 467-469

“Under 50% less water, the tomato cultivars Μ82 and Santorini were found to have no difference in photochemical efficiency compared to control plants and thus can be regarded as tolerant to water deficit.”

  1. 50% may be the optimum water content, and thus not a deficit. See others similar comments.

Lines 469-471

“Photosystem II efficiency of the wild tomato S. pennellii was the most tolerant to drought, as it was expected, and could manage to have an adequate photochemical function with almost 30% water regime of well-watered plants.”

  1. But are you comparing like with like. Why compare a wild species against two cvs.
  2. What is the effect of fruit load?
  3. What is the effect of water deficit on fruit production. After all, these plants are being grown, and their water optimized, for fruit cropping. In this respect, surely key data is missing, e.g. yield and fruit quality.

Lines 480-482

“Future extended experiments based on the evaluation of the redox state of QA can be applied to agricultural systems to reduce irrigation and increase productivity at far lower costs, helping to enhance agricultural sustainability under a global climate change.”

  1. The optimization of irrigation is routine for many producers. Please explain what is novel.

Lines 483-486

“Based on our results, we can propose deficit irrigation as an effective strategy for producing environmentally sustainable food, and apart from being a water-saving strategy, it is regarded as an important agronomic tradition that increases important primary and secondary metabolites that are essential for human health [33].”

  1. The optimization of irrigation is routine for many producers. Please explain what is novel.
  2. This sentence is contradictory.
  3. Why the emphasis, in this part of the ms., on primary and secondary metabolites. Moreover, other factors are key for human health, and there are many other considerations to take into account, across the value/supply chain.

Author Response

This is a well-structured, easily comprehended ms., describing the findings of what appears to be a competent piece of research. The methods and equipment used seem to be appropriate, given the experimentation requirements and questions at hand. I do, however, have a number of key questions and reservations, which I feel need to be carefully addressed. In particular, some of my comments draw question as to the uniqueness and strength of the reported outcomes and implications given in the paper; these require particular attention. Some of the comments that need to be addressed are based around climate understanding and are thus germane to the periodical of choice, title of the paper, and ms. narrative.

Thank you for your constructive comments that helped us to improve our MS.

Specific comments, by line(s)

Line 11

“Global climate change greatly influences plant growth and development, eventually affecting crop yield and quality”

  1. This needs rephrasing. Global climate change is not the ultimate driver but rather climate variables, via weather. The outcomes of climate change not only influences crop yield and quality (morbidity), but also survival (mortality).

The sentence was changed to “Fluctuations of the weather conditions, due to global climate change, greatly influence plant growth and development, eventually affecting crop yield and quality, but also plant survival.” (Lines 11-13).

Lines 26-28

“With almost 50% less water of well-watered plants, cv. Μ82 and cv. Santorini can be regarded, from all tomato cultivars examined, as water deficit-tolerant, with no difference in photochemical efficiency to control plants.”

  1. This doesn’t make any sense. Only two tomato cultivars were examined; and two were water deficit-tolerant. Does this study include enough cv. replicates to make the overarching claims made in the ms.

We changed the sentence to: “With 50% deficit irrigation cv. Μ82 and cv. Santorini did not show any difference in photochemical efficiency to control plants and are recommended to be cultivated under deficit irrigation as an effective strategy to enhance agricultural sustainability under a global climate change.” (Lines 28-31).

We examined four tomato cultivars and the wild species. As it was expected the wild species was found from the analysis of chlorophyll fluorescence parameters to be the most tolerant to water deficit. Experiments were repeated three times with 3-5 plants measured in each experiment under each treatment (Lines 191-192).

We also added in conclusion “Future extended experiments with more cultivars and growth stages based on the evaluation of the redox state of QA can be applied to agricultural systems to reduce irrigation….” (Lines 485-488).

  1. What about 30% less water scenario. This is not reported in the abstract and doesn’t get much coverage in the ms.

We included the 30% less water scenario in the abstract (Lines 26-28).

  1. 50% may be the optimum water content, and thus not a deficit. See comment directly below.

If 50% water irrigation was the optimum one, and not a deficit one, then all cultivars with 50% water irrigation would have optimum photochemical activity and agronomic traits.

Lines 28-29

“Thus, we can propose deficit irrigation as an effective strategy to enhance agricultural sustainability under a global climate change.”

  1. I don’t think this is actually deficit irrigation. Most irrigation schemes do not aim for field capacity, but rather what the plant requires for healthy growth and productivity.

We changed the sentence to: “With 50% deficit irrigation cv. Μ82 and cv. Santorini did not show any difference in photochemical efficiency to control plants and are recommended to be cultivated under deficit irrigation as an effective strategy to enhance agricultural sustainability under a global climate change.” (Lines 28-31).

Line 34

“As a consequence of global climate change the frequency, intensity, and duration of drought is increasing and has now reach[ed] to an alarming level [1-3]”

  1. Generally the English usage in the paper is very good, but please check ms. carefully and update accordingly.

Thank you for pointing this grammar mistake. We checked again the whole MS.

Lines 37-39

“Drought stress results from below-normal precipitation, frequently combined with warm temperatures, triggering widespread damages to plants and increased risk of wildfires [2].”

  1. There are also many other factors, as you have alluded to in lines 66-68. These should be brought together under the same section, and other factors itemized and briefly discussed.

We combined the two text parts (Lines 42-47).

Lines 46-47

“Drought-sensitive crops like the cultivated tomato (Solanum lycopersicum L.) are especially susceptible to impaired water availability due to climate change.”

  1. Why? Aren’t most crops susceptible?

Tomato is among the most water demanding species with 30-60 litters water per kg of tomato

  1. The emphasis is on water availability, but photosynthetic efficiency/performance is also strongly influence by temperatures.

Yes, temperature also strongly influences photosynthetic efficiency/performance and we mention it in lines 61-64.

Lines 47-50

“Solanum lycopersicum, is an annual species belonging to the family Solanaceae that originated in South America and now grown worldwide, as the most popular homegrown vegetable, for its edible fruits.”

  1. Please add reference.

References was added (Line 57).

Lines 66-68

“Although water deficit is the main cause for drought stress, increases in evapotranspiration under a warming climate are considered as the main cause for the extensive drying under global warming [2]”

Please see comment above for lines 37-39.

We combined the two text parts  (Lines 42-47) and we mentioned the influence of temperature in photosynthetic efficiency/performance (Lines 61-64).

Lines 68-70, and lines 70-73

“On the other hand, deficit irrigation, specifically using less water than the plant requirements, is proposed as an effective strategy for producing environmentally sustainable food [33].”

“Producing fruits and vegetables by deficit irrigation will contribute optimistically to the future of the planet and apart from being a water-saving strategy, deficit irrigation has become an important agronomic tradition to regulate many fruit quality variables [8].

  1. Most of the world’s food in grown under rain-fed conditions. Surely non-irrigation is the key, particularly given the carbon requirement needed to undertake irrigation, and the increasing demand for drinking water.
  2. “…will contribute optimistically to the future of the planet…”

I don’t think so. Reducing irrigation will help, but overall the impact would be negative’ unless the water was reclaimed using carbon-free technology.

We changed the sentence to: “Producing fruits and vegetables by deficit irrigation, apart from being a water-saving strategy, has become an important agronomic tradition to regulate many fruit quality variables [8].” (Lines 75-77).

  1. I think it’s also important to remember that much of the world’s commercial tomato production uses hydroponics, with already carefully monitored water usage.

Yes, it is true but we don’t now the percentage of world’s commercial tomato production with hydroponic systems. Yet, water used in hydroponic systems is about 450 ml/plant/day at full production. (A new method for hydroponic tomato production By Dario Stefanelli, Janine Jaeger, Rod Jone. Practical Hydroponics & Greenhouses, March 2013, 23. https://www.researchgate.net/publication/236736931_A_New_Method_for_Hydroponic_Tomato_Production).

Lines 115-117.

“All plants were grown in pots filled with peat (Terraplant, Compo) in a greenhouse at the Institute of Plant Breeding and Genetic Resources, (ELGO-Dimitra), Thermi, Thessaloniki, under 16h/8h photoperiod and temperature 22°C ± 3°C (day)/18 ± 3°C (night).”

  1. What were the growth stage used?
  2. Over what time period?
  3. Were all life stages included, and most importantly the fruiting stage?

We included all this information (Lines 124-133).

Lines 119-124

“In our experimental design we used plants from three watering regimes, determined by preliminary experiments, as follows: well-watered plants (control, 75 ± 2 % of field water capacity), moderate drought stressed (MoDS), with soil water status 55 ± 5% 121 of well-watered plants, whose watering stopped four days before sampling, and severe 122 drought stressed plants (SDS), with soil water status 32 ± 5% of well-watered plants,  whose watering stopped ten days before sampling.

  1. In addition to % of field capacity, also provide the actual readings from the probe/sensor in m3 m−3.
  2. The amount of water available to the plant also depends on the soil. Ideally, soil water potential should have been measured with a suitable Decagon probe.
  3. Ideally a further water deficit trial should have been included, as 30% deficit still provides the plant with sufficient water. This is borne out by the results.

Unfortunately, we have not measured soil water potential and have not included further water deficit trials.

Lines 331-333

“The maximum efficiency of PSII photochemistry (Fv/Fm) is among the chlorophyll fluorescence parameters mostly used to evaluate drought stress impact on plants and for selection of drought-tolerant varieties [25,74].”

  1. Cultivars, not varieties.

Yes, we changed it (Line 332).

Lines 467-469

“Under 50% less water, the tomato cultivars Μ82 and Santorini were found to have no difference in photochemical efficiency compared to control plants and thus can be regarded as tolerant to water deficit.”

  1. 50% may be the optimum water content, and thus not a deficit. See others similar comments.

As we have mentioned before if 50% water irrigation was the optimum one, and not a deficit one, then all cultivars with 50% water irrigation would have optimum photochemical activity and agronomic traits.

Lines 469-471

“Photosystem II efficiency of the wild tomato S. pennellii was the most tolerant to drought, as it was expected, and could manage to have an adequate photochemical function with almost 30% water regime of well-watered plants.”

  1. But are you comparing like with like. Why compare a wild species against two cvs.

We included Solanum pennellii  as a known drought tolerant species (Line 107) in order to compare the other 4 cultivars with it.

  1. What is the effect of fruit load?
  2. What is the effect of water deficit on fruit production. After all, these plants are being grown, and their water optimized, for fruit cropping. In this respect, surely key data is missing, e.g. yield and fruit quality.

We have not tested the effect of deficit irrigation on the fruit load. We know that this is important and thus we mention in the conclusion section (Lines 485-488) “Future extended experiments with more cultivars and growth stages based on the evaluation of the redox state of QA can be applied to agricultural systems to reduce irrigation….”

Lines 480-482

“Future extended experiments based on the evaluation of the redox state of QA can be applied to agricultural systems to reduce irrigation and increase productivity at far lower costs, helping to enhance agricultural sustainability under a global climate change.”

  1. The optimization of irrigation is routine for many producers. Please explain what is novel.

Yes, your comment touches on an important point that was missing. Thank you for pointing it. There are many phenotyping methods indeed but chlorophyll fluorescence analysis is a very quick method among the non-destructive phenotyping technologies. The novel finding of our research is that the previously proposed parameter for the evaluation of drought tolerance Fv/Fm is not a suitable one, since as it was shown recently it is not related to the efficiency of PSII photochemistry, but the redox state of quinone A (QA), as estimated by the chlorophyll fluorescence parameter 1−qL, as we show, is a good indicator to probe photosynthetic efficiency under drought stress (Lines 479-484).

Lines 483-486

“Based on our results, we can propose deficit irrigation as an effective strategy for producing environmentally sustainable food, and apart from being a water-saving strategy, it is regarded as an important agronomic tradition that increases important primary and secondary metabolites that are essential for human health [33].”

  1. The optimization of irrigation is routine for many producers. Please explain what is novel.

As we have mentioned before we added such information to substitute the sentence in Lines 483-486 (Lines 479-484).

  1. This sentence is contradictory.
  2. Why the emphasis, in this part of the ms., on primary and secondary metabolites. Moreover, other factors are key for human health, and there are many other considerations to take into account, across the value/supply chain.

This sentence has been deleted from conclusion section, modified and moved to introduction (Lines 73-79).

Reviewer 2 Report

The study of the paper "Harnessing Chlorophyll Fluorescence for Phenotyping Analysis of Wild and Cultivated Tomato for High Photochemical Efficiency under Water Deficit for Climate Change Resilience" focuses on an experiment to evaluate the efficacy of the use of Chlorophyll Fluorescence on Tomato performance under Water Deficit. Although the research is not particularly innovative, this study is interesting, and the experimental design and analysis were adequately carried out.

At current state, the manuscript is suitable for publication after minor revisions.

Minor comments/suggestions follow.

Line 73: Please, explain the fruit quality variables

Lines 299-306: Please, Substitute Regression for correlation. It would be advisable to insert the R2 values in the text and comment on the results

Lines 307-314: Please, Substitute Regression for correlation. It would be advisable to insert the R2 values in the text and comment on the results

Lines 385-387: Please, Substitute Regression coefficient for correlation coefficient

Lines 462-466: Please, remove (Figures 2b,4b,5b), (Figure 1d), (Figures 1a,b,c)

Lines 476-479: Please, remove this sentence. It is suitable for an introduction and not for conclusions

Lines 484-486: Please remove “and apart from being a water-saving strategy, it is regarded as an important agronomic tradition that increases important primary and secondary metabolites that are essential for human health [33]”. It could be appropriately moved in the introduction.

Author Response

The study of the paper "Harnessing Chlorophyll Fluorescence for Phenotyping Analysis of Wild and Cultivated Tomato for High Photochemical Efficiency under Water Deficit for Climate Change Resilience" focuses on an experiment to evaluate the efficacy of the use of Chlorophyll Fluorescence on Tomato performance under Water Deficit. Although the research is not particularly innovative, this study is interesting, and the experimental design and analysis were adequately carried out.

At current state, the manuscript is suitable for publication after minor revisions.

Thank you for your constructive comments that helped us to improve our MS.

Minor comments/suggestions follow.

Line 73: Please, explain the fruit quality variables.

We added this information (Line 77).

Lines 299-306: Please, Substitute Regression for correlation. It would be advisable to insert the R2 values in the text and comment on the results.

Lines 307-314: Please, Substitute Regression for correlation. It would be advisable to insert the R2 values in the text and comment on the results

Lines 385-387: Please, Substitute Regression coefficient for correlation coefficient

We substitute in all requested lines "regression" for "correlation" (Lines 298, 306, 384-385) and inserted R2 values in the text (Lines 302, 311,385-386).

Lines 462-466: Please, remove (Figures 2b,4b,5b), (Figure 1d), (Figures 1a,b,c).

We removed all these.

Lines 476-479: Please, remove this sentence. It is suitable for an introduction and not for conclusions.

We changed this sentence in order to be suitable for the conclusion section (Lines 475-478).

Lines 484-486: Please remove “and apart from being a water-saving strategy, it is regarded as an important agronomic tradition that increases important primary and secondary metabolites that are essential for human health [33]”. It could be appropriately moved in the introduction.

We removed this sentence to introduction (Lines 75-78).

Reviewer 3 Report

It is an interesting article. Authors are suggested to incorporate below mentioned suggestions:

Abstract: Nicely written.

Introduction: Authors should improve the hypothesis part. Additionally, different sections needs a better connection for a story like flow.

Which ANOVA was performed before Duncan's test? mention

Figure 1, mention units on the Y-axis, e.g. plant height (cm) etc.

Same for other figures. I understand units are mentioned in legends, but its better for readers if mentioned in y-axis

Discussion: needs to be improved by molecular point of view. Authors must discuss about molecular mechanisms as well.

Author Response

It is an interesting article. Authors are suggested to incorporate below mentioned suggestions:

Abstract: Nicely written.

Introduction: Authors should improve the hypothesis part. Additionally, different sections needs a better connection for a story like flow.

Thank you for your constructive comments that helped us to improve our MS. We rewritten the hypothesis part and rearranged some parts of the text.

Which ANOVA was performed before Duncan's test? Mention

We mentioned in the subsection of Statistical Analysis that we performed one-way analysis of variance (ANOVA).

Figure 1, mention units on the Y-axis, e.g. plant height (cm) etc.

Same for other figures. I understand units are mentioned in legends, but its better for readers if mentioned in y-axis

We did it.

Discussion: needs to be improved by molecular point of view. Authors must discuss about molecular mechanisms as well.

We discuss only photosynthetic responses to drought stress, but our discussion is based mainly on the molecular mechanisms of photosynthesis (e.g., the photoprotective mechanism of NPQ is a molecular mechanism of photoprotection).

Round 2

Reviewer 1 Report

Thank you for your careful attention to my comments. 

Reviewer 3 Report

Authors have addressed all of my comments.